# Methylated CpG ODNs from *Bifidobacterium longum* subsp. *infantis* Modulate Treg Induction and Suppress Allergic Response in a Murine Model

**DOI:** 10.3390/ijms26146755

**Published:** 2025-07-14

**Authors:** Dongmei Li, Idalia Cruz, Samantha N. Peltak, Patricia L. Foley, Joseph A. Bellanti

**Affiliations:** 1Department of Microbiology & Immunology, Georgetown University Medical Center, Washington, DC 20057, USA; dl33@georgetown.edu (D.L.);; 2Department of Oncology, Animal Models, Shared Resources, Georgetown University Medical Center, Washington, DC 20057, USA; 3Department of Pediatrics, Georgetown University Medical Center, Washington, DC 20057, USA

**Keywords:** methylated CpG, oligodeoxynucleotides, Treg, IL-10, OVA-allergic murine model

## Abstract

In our previous studies, methylated CpG oligodeoxynucleotides (ODN) derived from *Bifidobacterium longum* subsp. *infantis* have demonstrated immunomodulatory effects through the induction of regulatory T cells (Tregs). To define the structural determinants underlying this effect, we synthesized four CpG ODNs varying in methylation degree, CpG motif placement, and backbone length. These include (1) ODN-A (2m-V1), a 20-nucleotide CpG oligodeoxynucleotide incorporating two 5-methylcytosines at positions 4 and 12 within centrally placed CpG motifs; (2) ODN-B (um-V2), a 20-nucleotide CpG oligodeoxynucleotide with a backbone structure identical to ODN-A but unmethylated; (3) ODN-C (2m’-V3), a 20-nucleotide CpG oligodeoxynucleotide with a backbone structure identical to ODN-A, but with two 5-methylcytosines shifted to positions 7 and 15; (4) ODN-D (3m-V4), a 27-nucleotide CpG oligodeoxynucleotide with an extended backbone structure, this time with three 5-methylcytosines at positions 3, 11, and 19. Using a murine model of an OVA-induced allergy, we show that methylated ODN-A (2m-V1) and ODN-D (3m-V4) markedly reduce serum anti-OVA IgE, clinical symptoms, eosinophilic infiltration, and Th2/Th17 responses, while promoting splenic Treg expansion and IL-10 production. In contrast, unmethylated ODN-B (um-V2) and a positionally altered methylated ODN-C (2m’-V3) both failed to suppress allergic inflammation, and, in contrast, enhanced the Th2/Th17 response and induced robust in vitro Toll-like receptors TLR7/8/9 expression in native splenocytes. These findings suggest that both methylation and motif architecture critically influence the immunologic profile of CpG ODNs. Our results provide mechanistic insights into CpG ODN structure/function relationships and support the therapeutic potential of select methylated sequences for restoring immune tolerance in allergic diseases.

## 1. Introduction

Among all the available treatments for allergic diseases, allergen-specific immunotherapy (AIT) uniquely offers long-term efficacy by modifying the underlying immune response. The immunomodulatory properties of DNA, particularly synthetic and microbially derived oligodeoxynucleotides (ODNs) used as immune conjugates, have garnered increasing interest as a strategy to treat a wide variety of diseases, including cancers, infections [1,2], and recent allergic diseases [3,4]. Among these, CpG-containing ODNs are of particular interest in that they can be used either as a standalone molecule or as an adjuvant to modulate innate and adaptive immunity through Toll-like receptor 9 (TLR9) signaling [5,6]. In the context of allergic diseases, current unmethylated CpG ODN adjuvants are designed to shift Th2-skewed allergic responses toward a Th1-biased profile to reduce allergies, but this immune activation may also increase TNF-α levels that have been associated with toxic shocks in mice [7,8]. However, while unmethylated CpG motifs typically promote immune activation and are being developed as vaccine adjuvants for cancer immunotherapy [9,10], certain CpG ODNs, such as the TLR-9 antagonist CpG-c41 [11,12], exert immunosuppressive or tolerance-inducing effects. Apparently, influences on innate and adaptive immunity depend, to some degree, on CpG-containing DNA sequences, methylation statuses, and epigenetic structural features [5,13,14].

Using CpG ODN therapies to restore immune tolerance in allergic and autoimmune diseases is the primary focus of our work [15]. Building on this growing body of evidence, our group has reported that genomic DNA from *Bifidobacterium longum* subsp. *Infantis* selectively induces Foxp3^+^ regulatory T cells (Tregs) from human peripheral blood mononuclear cells in vitro [16]. In the same study, methylome sequencing identified a unique methylated m5C-containing CpG motif in this probiotic strain. A synthetic CpG ODN derived from this methylated motif demonstrated potent Treg-inducing activity in vitro. These findings were further validated in vivo using a murine model of allergic inflammation, where administration of the methylated CpG ODN suppressed inflammatory symptoms, reduced Th2 cytokine levels in serum, decreased mast cell infiltration at the injection site, and enhanced the differentiation of splenic Foxp3^+^ Tregs [17]. A subsequent dose–response study confirmed that the Treg-promoting effect was indeed dose-dependent and preferentially associated with the methylated, rather than the unmethylated, form of the ODN [18]. Together, these findings highlight the therapeutic potential of probiotic-derived methylated CpG motifs for modulating immune responses in both hypersensitivity and systemic lupus erythematosus (SLE) [19].

While these results support the therapeutic potential of methylated CpG ODNs, the molecular basis for their differential activity remains undefined. In particular, the contribution of specific structural elements—such as the number and position of methyl groups, backbone length, and CpG spacing—to their immunoregulatory function has not been systematically studied.

In the present work, we investigate the structure/function relationships of a panel of synthetic CpG ODN variants derived from the native *Bifidobacterium longum* subsp. *infantis* sequence. We hypothesize that defined modifications in methylation and backbone architecture will differentially impact their ability to induce Foxp3^+^ Tregs and modulate allergic inflammation. Using a murine model of ovalbumin (OVA) allergy, we analyzed immunologic endpoints including Treg differentiation, cytokine profiles, serum-specific anti-IgE level, and histopathologic inflammation.

Our results identify key structural determinants of CpG ODN-mediated immune regulations and offer mechanistic insights into the rational design of nucleic-acid-based immunotherapies for allergic disease.

## 2. Results

To evaluate the immunologic consequences of structural modifications in CpG ODNs, we tested four synthetic variants that differed in methylation status, CpG motif positioning, and backbone length (Figure 1A). For the ODN nomenclature, the prefix number indicates the number of methylated cytosine residues. For example, “2m” in 2m-V1 and “3m” in 3m-V4 correspond to two and three methylated cytosine(s), respectively, while “um” in um-V2 refers to an unmethylated ODN. The designation 2m’-V3 indicates two methylated cytosine(s) located at alternative positions compared to a naïve *Bifidobacterium* methylated motif. The “V” denotes “vaccine,” and the accompanying number reflects the chronological order in which the variants were designed. A murine model of ovalbumin (OVA)-induced allergic inflammation was used to assess the in vivo immunological effects of co-treatment with each CpG ODN variant (Figure 1B). Key outcome measures included serum IgE levels, clinical allergy scores, tissue histopathology, frequencies of regulatory T cells (Tregs) and effecter T cells (Teff), cytokine profiles, and expression levels of nucleic-acid-sensing Toll-like receptors (TLRs).

### 2.1. Methylated 3m-V4 and 2m-V1 Suppress Serum Anti-OVA IgE Levels and Allergic Symptoms

Serum levels of OVA-specific IgE were measured on Day 22, 24 h after the OVA challenge. As expected, the OVA challenge after two weeks of sensitization significantly elevated anti-OVA IgE levels in mouse sera. However, this increase was effectively suppressed in mice when co-treated with methylated 3m-V4 and 2m-V1 (Figure 2A). In contrast, mice receiving unmethylated um-V2 or positionally altered methylated 2m’-V3 maintained high levels of serum IgE.

Consistent with the elevated IgE levels, allergic symptom scores at a day earlier (1 h post the OVA challenge on Day 21) varied in different ODN-co-treated groups. After a 200 µg OVA challenge, the severity of allergic symptoms, measured by the frequency of scratching (using front or rear legs) and scratching area (ear and or subcutaneously injected back), was significantly higher in the OVA-alone and OVA + um-V2 treated mice (Figure 2B). These allergic symptoms were less manifested in the mouse groups receiving the co-treatment with methylated ODNs, particularly 2m-V1 and 3m-V4.

### 2.2. Methylated 3m-V4 and 2m-V1 Partially or Fully Restore the OVA-Induced CD4^+^ Reduction

On Day 22, the proportion of viable splenocytes in the vehicle control group (PBS–AddaVax-treated mice) remained high at 85.26%, with only minor changes observed across most treatment groups (Figure 3A). In contrast, OVA-challenged mice without any CpG ODN treatment (ova/wo) exhibited a significant reduction in splenocyte viability, dropping to 81.86% (*p* < 0.05). This decline was mitigated by the co-administration of any of the four CpG ODN variants tested. Among them, 3m-V4 and 2m-V1 demonstrated a stronger protective effect compared to the unmethylated um-V2 and the positionally methylated 2m’-V3. Notably, there were no statistically significant differences in splenocyte viability between any of the OVA+ODN treatment groups and the PBS control, suggesting that these CpG ODNs not only lack cytotoxic effects but may also confer a degree of protection against OVA-induced immune cell stress.

Different from cell viability, the CD4^+^ T-cell population in the spleen varied depending on the specific ODN used (Figure 3B). Consistent with reduced cell viability, CD4^+^ cell frequency was significantly decreased in OVA-challenged mice compared to controls (*p* < 0.001). Co-injection with 2m-V1 or 3m-V4 effectively prevented this reduction, showing statistically significant restoration of CD4^+^ cells when compared to the ova/wo group (*p* < 0.05 and *p* < 0.0001, respectively). In contrast, unmethylated um-V2 and positionally methylated 2m’-V3 failed to restore CD4^+^ cell levels, indicating that their ability to counteract OVA-induced CD4^+^ reduction was limited.

### 2.3. Methylated 3m-V4 and 2m-V1 Increase Treg Cells and Reduce Th2 and Th17 Subsets in OVA-Challenged Mice

Despite the overall reduction in the CD4^+^ T-cell population, the percentage of Treg cells in OVA-challenged mice was slightly higher than in the negative control group, though this difference was not statistically significant. Co-administration of each ODN with OVA further increased the proportion of splenic Treg cells (CD25^+^FOXP3^+^) (Figure 4A), including significant increases in the um-V2 (*p* = 0.0025) and 2m’-V3 (*p* = 0.0182) groups compared to the PBS control. Importantly, when compared to the OVA-challenged group, Treg percentages were significantly elevated in mice treated with 3m-V4 (*p* = 0.023) and 2m-V1 (*p* = 0.01). Together with their higher total CD4^+^-cell counts, these findings strongly support a Treg-promoting effect of 3m-V4 and 2m-V1 in OVA-challenged mice.

To examine how this regulatory T-cell response aligns with effecter T-cell (Teff) activity, we analyzed Th2 (GATA3^+^IL-4^+^), Th17 (CD4^+^IL-17^+^), and Th1 (T-bet^+^IFN-γ^+^) populations as well. As shown in Figure 4B, OVA-challenged mice exhibited a pronounced increase in Th2 cells. This elevation was significantly suppressed by co-treatment with methylated 3m-V4 (*p* = 0.0039), while 2m-V1 showed a similar trend that did not reach significance. In contrast, co-treatment with unmethylated um-V2 significantly increased Th2 frequency compared to the OVA-only group (*p* = 0.0017), suggesting a possible exacerbation of Th2 inflammation. A slight, non-significant increase in Th2 cells was also observed in the 2m’-V3 group. Regarding the Th17 response (Figure 4C), unmethylated um-V2 was the only condition that significantly elevated Th17 levels compared to OVA alone mice. In contrast, the Th1 population (Figure 4D) remained unchanged across all treatment groups, including PBS controls, OVA-challenged mice, and those co-treated with various ODNs.

The overall immune responses in each group were then evaluated by their Treg/Teff ratios. The Treg/Th2 ratio was significantly decreased in OVA-only mice compared to the PBS control (*p* = 0.034) as expected, indicating a shift toward pro-allergic inflammation. This imbalance was effectively corrected by 3m-V4 co-treatment (*p* = 0.006). In contrast, the elevated Th2 responses in um-V2 and 2m’-V3 groups maintained a lower Treg/Th2 ratio. Compared with the Treg/T2 ratio, the Treg/Th17 ratio was less affected in OVA-only mice, but a pronounced reduction was observed in the OVA/um-V2 group, which showed elevated Th17 levels. Remarkably, the 3m-V4 group displayed both higher Treg levels and significantly suppressed Th17 responses, demonstrated by its significantly higher Treg/Th17 ratio compared to OVA-only mice.

Taken together, these results suggest that 3m-V4, and to a lesser extent 2m-V1, can counteract OVA-induced Th2- and Th17-driven inflammation by promoting a strong Treg response. In contrast, unmethylated um-V2 and positionally modified 2m’-V3 either failed to restore the immune balance or exacerbated the pro-inflammatory environment, thereby disrupting the equilibrium between Treg and effecter T-cell subsets.

### 2.4. Elevated IL-5 and IL-4 in um-V2 and 2m’-V3 Co-Treated Mouse Sera Contrast to IL-10 Elevation in 3m-V4 and 2m-V1 Mice

Among the eight cytokines analyzed using the LEGENDplex™ Multi-Analyte Flow Assay Kit, IL-13 and IL-2 showed no significant differences between the treatment groups and the negative control. The remaining six cytokines varied depending on the treatment group.

As shown in the top panel of Figure 5A, the Th2-related cytokine IL-4 was elevated in OVA-only mice compared to negative controls and further increased in the 2m’-V3 co-treated group. IL-5 levels were elevated in all OVA-treated groups, regardless of ODN co-treatment. However, co-treatment with unmethylated um-V2 or positionally altered methylated 2m’-V3 significantly increased serum IL-5 compared to PBS controls. The co-treatment with um-V2 showed the most pronounced effect, highlighted by significantly higher IL-5 levels in this group when compared to the OVA-only group (*p* = 0.001). Similarly, the Th17-associated cytokine IL-6 was significantly elevated in the OVA + um-V2 group (*p* = 0.01), suggesting a potential synergistic inflammatory response driven by unmethylated um-V2.

Alongside less-Th1-cell phenotypes observed in the spleen, Th1-related cytokines were also affected. We observed that TNF-α in serum remained unchanged across all groups. However, the IFN-γ level was significantly elevated in the OVA + 3m-V4 group compared to the negative control (*p* = 0.0021). A similar pattern was observed for IL-10 (Figure 5A, the bottom panel), which was also significantly increased in the OVA + 3m-V4 group (*p* = 0.017).

To validate the anti-inflammatory potential of methylated ODNs, we used ELISA to measure IL-10 across all treatment groups and TGF-β levels between the 3M-V4 and um-V2 conditions. As shown in Figure 5B, IL-10 levels were significantly elevated in OVA-challenged mice co-treated with methylated 2m-V1 and 3m-V4, with *p*-values of 0.03 and 0.01, respectively, compared to the OVA-only group. Consistent with our previous findings [18], TGF-β levels were not universally increased in response to OVA or ODN challenges, and neither um-V2 nor 3m-V4 altered this OVA-induced baseline.

Taken together, these cytokine profiles indicate that methylated ODNs, particularly 3m-V4, promote anti-inflammatory cytokine responses, potentially counteracting OVA-induced inflammation. Instead, unmethylated um-V2 and the positionally altered methylated 2m’-V3 appear to exacerbate the inflammatory response.

### 2.5. OVA-Induced Local Skin Inflammation Is Exacerbated by Unmethylated um-V2 and Alternate Methylated 2m’-V3 with Elevated Serum IL-5 Response

Histopathologic changes at the injection site were evaluated by H&E staining. Inflammatory cell infiltration in the dermis was clearly observed in OVA-treated mice and was more pronounced in those co-treated with um-V2 or 2m’-V3 (Figure 6A). Compared to the OVA-alone group, where inflammation was more diffusely distributed throughout the dermal layer, the co-treatment with um-V2 or 2m’-V3 resulted in a more compact and denser band of inflammatory cells.

Two predominant cell types were identified within the infiltrates. The first type consisted of small, round cells with dense, darkly stained nuclei and minimal cytoplasm, consistent with lymphocytes located between adipose tissue (white vacuolated cells) and skeletal muscle (pink striated fibers). The second cell type, eosinophils, exhibited distinct morphological features, including eosinophilic granules, bilobed nuclei, and red-stained cytoplasm, especially visible under Giemsa staining (Figure 6B). Notably, eosinophils were only abundantly distributed within the inflamed areas in the OVA + um-V2 and OVA + 2m’-V3 groups (indicated by red arrows). Manual quantification using ImageJ revealed a significant increase in the percentage of eosinophils within the infiltrate in the OVA + um-V2 and OVA + 2m’-V3 groups when compared to the negative control, OVA-alone, and the other two methylated ODN groups (Figure 6C). Notably, eosinophil percentages closely mirrored the serum IL-5 levels observed in each group, suggesting that elevated IL-5 expression drives these eosinophilic infiltrations.

### 2.6. Unmethylated um-V2 Strongly Induces Nucleic-Acid-Sensing TLRs In Vitro Mouse Splenocytes

To explore potential mechanisms underlying the differential immunological effects of CpG ODNs, we examined the expression of nucleic-acid-sensing Toll-like receptors (TLRs) in mouse splenocytes in vitro under ODNs treatment (Figure 7). Specifically, we assessed TLR7 and TLR8, which recognize single-stranded RNA, and TLR9, which detects DNA. Splenocytes isolated from a single untreated BALB/c mouse were cultured with 25 µg of unmethylated uM-V2, methylated 2m-V1, or alternate methylated 2m’-V3 for 2 and 6 h. RNA was extracted and compared with PBS-treated controls.

At the two-hour time point, methylated 2m-V1 induced a ~15-fold increase in TLR7 expression compared to PBS. However, by 6 h, splenocytes treated with unmethylated um-V2 showed a markedly elevated expression of all three TLRs, with nearly a 50-fold increase in TLR9, a 4.5-fold increase in TLR8, and a 4-fold increase in TLR7. These levels were significantly higher than those observed in any other treatment group. In contrast, both methylated 2m-V1 and 2m’-V3 did not induce notable changes in TLR expression at this time point. These expressional data suggest that unmethylated um-V2 robustly activate nucleic-acid-sensing TLRs, particularly TLR9, while methylated CpG ODNs may evade detection through these canonical pathways.

## 3. Discussion

This study demonstrates that the methylation status and sequence type of CpG ODNs profoundly influence allergic responses in an OVA-induced mouse model. Among the tested sequences, methylated 3m-V4 and 2m-V1 ODNs consistently exhibited immunomodulatory effects that attenuate Th2/Th17-driven allergic inflammation.

First, both methylated 3m-V4 and 2m-V1 significantly suppressed serum anti-OVA IgE levels and mitigated allergic symptoms, indicating a protective role in IgE-mediated hypersensitivity. This suppression aligns with their capacity to counteract the OVA-induced reduction in CD4^+^ T cells, suggesting a broader immune-restorative function. Importantly, the increase in CD4^+^CD25^+^Foxp3^+^ Treg cells, coupled with reductions in Th2 and Th17 subsets, points to a shift toward immune tolerance. This shift is likely mediated through regulatory circuits that counterbalance the pro-inflammatory milieu typically driven by IL-4 and IL-5 in allergic contexts.

Consistent with this, serum cytokine profiling revealed elevated IL-10 levels in 3m-V4- and 2m-V1-treated mice, supporting their anti-inflammatory effect. In contrast, co-administration of um-V2 or methylated 2m’-V3 intensified the allergic response, as evidenced by increased IL-4, IL-5, and eosinophilic infiltration at the injection site. These contrasting profiles suggest that not all CpG ODNs are beneficial in allergy modulation and that sequence-specific and methylation-dependent factors critically shape their outcomes.

The marked local skin inflammation observed in the um-V2 and 2m’-V3 groups further reinforces their pro-allergic potential. The associated eosinophil accumulation is consistent with the elevated IL-5 levels, underscoring a Th2-skewed response by them [20]. In contrast, the immunosuppressive effect of 3m-V4 and 2m-V1 may reflect a context-dependent induction of regulatory pathways, which is well known to override the allergen-induced immune deviation [21,22,23].

To further explore the underlying mechanisms, we examined the expression of nucleotide-sensing Toll-like receptors (TLRs) in splenocytes following in vitro ODN exposure. Unmethylated DNA activates TLR9, which is primarily known for its pro-inflammatory role [24,25]. However, recent research highlights TLR9’s capacity to dampen inflammation in certain contexts. For example, TLR9 and a downstream target, IFN-α, are required for the migration of myeloid-derived suppressor cells into gastric tissue to suppress the establishment of persistent *H. pylori* infections [26]. While methylated 2m-V1 induced minimal TLR expression in this study, unmethylated um-V2 triggered a striking upregulation of TLR9 (~50-fold), TLR8 (~4.5-fold), and TLR7 (~4-fold) at 6 h. These results, similar to others [24,25], suggest that unmethylated ODNs robustly engage canonical nucleic-acid-sensing pathways, particularly TLR9, which may also account for their pro-inflammatory and Th2-skewing effects observed in our in vivo model. Conversely, the limited TLR induction by methylated ODNs implies alternative recognition pathways, such as self-DNA sensing the cGAS-STING pathway [27] or reduced engagement with TLR-expressing immune subsets. Although these in vitro data are preliminary, this differential TLR engagement may contribute to the distinct immunological profiles observed in vivo.

The natural origin of this set of CpG ODNs from a probiotic bacterial genome supports their potential safety for use in human allergic disease. The differential immunomodulatory effects observed among the four ODN variants in this study clearly underscore the importance of both the methylation status and the precise positioning of methylated cytosine(s) in the design of anti-allergic DNA vaccines. The opposing roles of 2m-V1 and um-V2 on Treg induction, previously reported, were reaffirmed in this study. Furthermore, the inclusion of two newly tested methylated variants—2m’-V3 and 3m-V4—provided additional insights. Notably, 2m’-V3, despite containing two methylated cytosine(s) and having similar structural stability to 2m-V1, exhibited a Treg/Teff response pattern more closely resembling that of the unmethylated um-V2. This suggests that the precise position of methylation in 2m-V1 is critical for its immunomodulatory effect. Moreover, 3m-V4, which contains three methylated sites, demonstrated superior anti-allergic efficacy compared to 2m-V1, offering a promising example of how methylation density and placement can be optimized to enhance therapeutic outcomes. These findings provide a valuable framework for refining the design of methylated ODN-based vaccines by strategically tailoring their methylation patterns.

In summary, our findings support the therapeutic potential of selected methylated CpG ODNs, especially 3m-V4 and 2m-V1, in mitigating allergic inflammation. By dampening IgE responses, enhancing Treg populations, and avoiding overactivation of nucleic-acid-sensing TLRs, these ODNs may help restore immune balance. Further studies are warranted to dissect the TLR-independent mechanisms potentially activated by methylated DNA sequences and to assess their translational relevance in human allergy models.

## 4. Materials and Methods

### 4.1. Synthetic Short-Chain Oligodeoxynucleotides (ODNs), Allergen OVA, and Immunoadjuvant

Four synthetic ODNs were used in this study, which were synthesized by Integrated DNA Technologies (Coralville, IA, USA). As shown in Figure 1A, the sequence 2m-V1 is 20 bp in length, with the DNA sequence 5’-C*A*G*/iMe-dC/GGCGCCG/iMe-dC/GGCGC*C*T*G containing a duplicate of the Bl-T2 m5C motif [(A/G) GCpGGCGCC)] of *B. longum* subsp. *infantis* [16]. The unmethylated counterpart, um-V2 (5’-C*A*G*CGGCGCCGCGGCGC*C*T*G), corresponds to 2m-V1 without methylation. The new synthesized 2m’-V3 shares the same sequence as 2m-V1 but with different methylated CpG locations, and 3m-V4 is designed to contain a triplicate of the B1-T2 motif (Figure 1). All four ODNs feature a nuclease-resistant phosphorothioate backbone in the first and last three nucleotides, as indicated by “*”in Figure 1A.

Lyophilized ODNs were dissolved in PBS at 3 mg/mL and kept at −20 °C until use. The predicted secondary structure stability, accessed via Gibbs free energy (ΔG), revealed ΔG values of −11.3 kcal/mol for both 2m-V1 and 2m’-V3 and a less negative ΔG of −9.3 kcal/mol for um-V2, suggesting reduced structural stability in the unmethylated form. 3m-V4 is an extended version of the BI-T2 motif, consisting of 27 nucleotides with three methylated cytosine(s). Its ΔG value, calculated without phosphorothioate (PTO) and methylation modifications, was −20.3 kcal/mol. When PTO and methylation were included, ΔG further decreased from −21.8 to −22.8 kcal/mol, indicating the highest thermodynamic stability among the four ODNs.

The immunogenic low-endotoxic OVA (ovalbumin) was purchased from Chondrex, Inc. (Woodinville, WA, USA) with specified endotoxic levels < 1 EU/mg. A stock solution was made by dissolving 10 mg of the lyophilized form in 1 mL of sterile PBS and storing it at −20 °C until use. An immune adjuvant, AddaVax, was purchased from InvivoGen (SanDiego, CA, USA). This ready-to-use adjuvant was a squalene-based nanoemulsion and was stored at 4 °C prior to use. We mixed the AddaVax with different doses of the OVA, with and without ODN, pre-prepared in PBS at a 1:2 ratio. An aliquot of 100 µL of the mixture per mouse was then sterilized by filtering it through a 0.2 µm filter just before subcutaneous injection.

### 4.2. Animals and Groups

Female BALB/c mice aged 6–8 weeks were sourced from Jackson Laboratory (Bar Harbor, ME, USA) and housed in autoclaved plastic cages on individually ventilated racks at 70–74 °F in Georgetown University Division of Comparative Medicine (DCM). Mice were fed a 5053-PicoLab Rodent Diet 20 (LabDiet, St. Louis, MO, USA) that contained 20% protein without peanut or egg protein. The procurement occurred one week before the initiation of the experiment, allowing the mice to acclimatize within the DCM environment. During this period and throughout the entire experiment, the mice had access to food and water ad libitum, residing in a ventilated caging cabinet (<5) with environmental conditions including a 12 h light–dark cycle and humidity levels between 60 and 80%. Stringent ethical standards, in accordance with guidelines set forth by the Association for Assessment and Accreditation of Laboratory Animal Care International (AAALAC) and the Institutional Animal Care and Use Committee (IACUC) of Georgetown (Protocol #2022-0021), were strictly adhered to throughout the study.

A total of 30 mice were organized into six groups (A through F), with 6 mice in Groups B through E, while Groups A (PBS + AddaVax vehicle control) and F had only 3 mice each (Group F was the positive control). Group B mice were subcutaneously injected with OVA only (allergic condition induction control), and oligodeoxynucleotide-treated Groups C–F were treated, respectively, with an additional 2m’-V3, 3m-V4, um-V2, and 2m-V1.

Groups A and F comprised fewer animals (n = 3) to reduce overall animal use, as they have served in our experience as well-established controls in earlier experiments [17,18]. On the other hand, Groups C and D involved novel methylated ODN formulations (2m’-V3 and 3m-V4) and, therefore, required larger group sizes (n = 6) to evaluate variability and treatment effects. Group E also required 6 mice, based on earlier findings showing that the unmethylated um-V2 ODN consistently exacerbated allergic responses and reduced Treg levels, in contrast to methylated ODNs. Keeping this group equal in number to the other ODN-treated groups allowed us to assess the relative effectiveness of the new treatments with greater confidence.

### 4.3. OVA Sensitization and Challenge Model

An allergic model followed a previous report [17], and the injections were scheduled as in Figure 1B. Briefly, allergen OVA in 20 µg/mouse was introduced in Day 1 and Day 8 to sensitize the mice, except for the mice in Group A, and 200 µg on Day 21 to challenge the mice. OVA alone or with 100 µg ODN in PBS was mixed with an adjuvant, AddVax, in a 2:1 ratio to be subcutaneously injected in the middle part of the back area that had been shaved prior to the first injection. This mixture was prepared fresh on injection day and passed through a 0.2 µm syringe filter to ensure it was sterile. On Day −1, baseline blood samples were drawn from three mice in each group for measurement of serum IgE concentrations. The clinical symptoms observed within 1 h after antigen challenges were recorded, and all the mice were euthanized on Day 22. Blood and spleen specimens were collected, and 1.5 cm × 1.5 cm skin biopsy specimens were taken at the injection site.

### 4.4. Measurement of Serum Anti-OVA-IgE and Clinical Symptom

Serum concentrations of anti-OVA-specific IgE were determined using a mouse anti-OVA IgE antibody ELISA kit (Catalog #3004; Chondrex Inc., Woodinville, WA, USA), following the manufacturer’s instructions. Clinical symptoms were assessed 1 h after the OVA challenge on Day 21, using a scoring system, as previously described [17], as follows: 0: No evident symptoms; 1–2: Mild to intermediate symptoms (e.g., scratching and rubbing of the nose and head or intention on the injected site on the back, with fore limbs or hind limbs, edema around the eyes and mouth); 3: Severe symptoms (e.g., labored breathing, tremors, or major respiratory distress).

### 4.5. Histopathology and Eosinophils Percentage

Skin tissues from the injection sites were collected, fixed in 10% formalin, embedded in paraffin, and sectioned at 5 µm for hematoxylin and eosin (H&E) or Giemsa staining. Stained sections were scanned and evaluated for the severity of inflammation. Eosinophilic inflammation was also quantified by counting eosinophils within the inflammatory infiltration area, using skin samples from three mice per group.

Eosinophil quantification was performed by examining five 40× fields per section. In each field, the percentage of eosinophils was calculated relative to the total number of infiltrating cells (excluding epithelial cells) in the ImageJ software version 1.54. These percentages were then compared across treatment groups to assess differences in eosinophilic inflammation.

### 4.6. T-Cell Phenotypes in Spleen by Flow Cytometric Analysis

On Day 22, spleens were harvested from the mice, homogenized, and passed through a 40 µm cell strainer to prepare single-cell suspensions. Red blood cells (RBCs) were lysed using 3 mL of a cold RBC Lysis Buffer (Thermo Fisher Scientific, Waltham, MA, USA). Splenocytes were then washed with a cold complete RPMI medium by centrifugation at 1600 rpm for 10 min at 4 °C. Cell viability was assessed using 0.2% trypan blue.

T-cell subsets within the splenocyte population were identified using 10 fluorochrome-conjugated antibodies. For surface staining, 2 × 10^6^ splenocytes per well were plated in 96-well plates in 200 µL aliquots. After two washes, cells were stained with Zombie R718 viability dye (BioLegend, San Diego, CA, USA; 1:500 dilution) for 15 min at room temperature. Surface staining was performed using fluorochrome-conjugated anti-mouse antibodies (BioLegend), including CD3BV605 (clone 17A2), CD4BV785 (clone GK1.5), and CD25APC-Cy7 (clone PC61.5).

For intracellular staining, cells were fixed and permeabilized using the Intracellular Staining Fixation Buffer and Perm/Wash Buffer set (BioLegend). Intracellular markers were stained in Perm/Wash Buffer using the following fluorochrome-conjugated antibodies (BioLegend), including anti-Tbet BV421 (clone 4B10), GATA3 PerCPCY5.5 (clone 16E10A23), Foxp3 AF488 (clone MF-14), IFN-γ PE/Dazzle 594 (clone XMG1.2), IL-4 BV711 (clone 11B11/BVD6-24G2), IL-17APE (clone TC11-18H10.1), and IL-10 APC (clone JES5-16E3). Flow cytometric analysis was performed using a BD Fortessa SORP (BD Biosciences, Franklin Lakes, NJ, USA), and data were analyzed with FCS Express 7 (DeNovo Software, Pasadena, CA, USA). Each fluorochrome-conjugated antibody was compensated using Compensation Beads (BioLegend). Fluorescence Minus One (FMO) controls were included for each antibody in every experiment. Treg cells (CD25^+^Foxp3^+^) and Th1 (IFNγ^+^T-bet^+^), Th2 (IL-4^+^GATA3^+^), and Th17 (IL-17A^+^CD4^+^) populations were gated from viable CD3^+^CD4^+^ T cells, as outlined in the gating strategy shown in Appendix A.

### 4.7. Serum Cytokine Responses in ODN-Treatedmice

Mouse serum was collected from whole blood by centrifugation at 2000 rpm for 10 min at 4 °C. Serum samples were stored at −80 °C until use. Cytokine concentrations were measured using the LEGENDplex™Multi-Analyte Flow Assay Kit (BioLegend), specifically the mouse Th1/Th2 Panel (8-Plex), following the manufacturer’s instructions. The panel included IFN-γ, IL-5, TNF-α, IL-2, IL-4, IL-6, IL-10, and IL-13. Total TGF-β1 and IL-10 levels were additionally quantified using the BioLegend MAX™ Total TGF-β1 ELISA Kit and IL-10 ELISA Kit, respectively.

### 4.8. In Vitro Splenocyte TLR7, TLR8, and TLR9 Transcriptions

Two million splenocytes isolated from PBS-control mice (Group A) were washed twice with a complete RPMI 1640 medium and resuspended in 0.5 mL medium in a 24-well plate. Cells were treated with 25 μg of 2m-V1 (methylated oligo), um-V2 (unmethylated oligo), or 2m’-V3 (methylated oligo) dissolved in 50 μL PBS. Control wells received 50 μL PBS without ODNs. Plates were incubated at 37 °C in a 5% CO_2_ atmosphere for 2 or 6 h. Treated splenocytes were harvested at 2 h or 6 h for RNA extraction using a TRIzol reagent (Invitrogen, Carlsbad, CA, USA). RNA purity was confirmed by an A260/A280 ratio between 1.8 and 2.0. Reverse transcription was performed using the High-Capacity cDNA Reverse Transcription Kit (Applied Biosystems, Waltham, MA, USA). Quantitative PCR was conducted using a SYBR Green PCR Master Mix (Applied Biosystems) to assess the expression of TLR7, TLR8, and TLR9, with the following primer sets: TLR7 Forward (5′-CATCTGTCTTTGCCCCTCTAAG) and Reverse (5′-AGTTTCTTGACCTGTGCCTC); TLR8 Forward (5′-GAGTTCCTTGAGCTGTTTGC) and Reverse (5′-ACTTCCTTTTCTGTGTACCCC); TLR9 Forward (5′-AGAGGAAGAGAAAGTGGGAGAG) and Reverse (5′-AGGACACAGAACCAGGACTAG); and internal control primers for MGADPH Forward (5′-TTCACCACCATGGAGAAGGC) and MGADPH Reverse (5’-AGTGATGGCATGGACTGTGG).

### 4.9. Statistical Analysis

Continuous variables are expressed as the means ± standard deviation (SD), and categorical data are presented as frequencies or percentages. We performed the Student’s *t*-test (comparing the two groups) and one-way ANOVA (comparing three or more groups), followed by Tukey’s (comparing all groups) or Dunnett’s (comparing treatments vs. control) post-tests to assess the significance of changes relative to controls as indicated. All statistical analyses used 2-sided tests, and a *p*-value ≤ 0.05 was considered significant. All tests were performed with GraphPad Prism10.0 (San Diego, CA, USA).

## Figures and Tables

**Figure 1 ijms-26-06755-f001:**
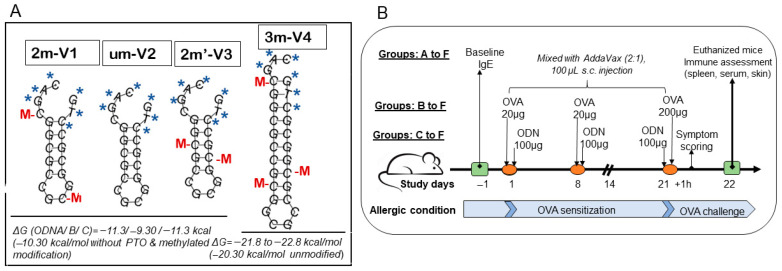
Secondary structures of four CpG oligodeoxynucleotides (ODNs) and experimental timeline for mouse allergy model. (**A**) The four CpG ODNs were designed based on the methylated cytosine-containing BI-T2 motif (RGCGGCGCC) derived from *Bifidobacterium longum* subsp. *Infantis* [16]. 2m-V1 is a 20 mer sequence containing two methylated cytosine(s) (red M) located according to the native m5C motif. um-V2 is the unmethylated version of 2m-V1, while 2m’-V3 includes methylated cytosine(s) at alternate positions. All three share identical base sequences but differ in methylation pattern. 3m-V4 is an extended version of the BI-T2 motif, comprising 27 bases with three methylated cytosine(s). “M” indicates methylated cytosine residues, and “*” denotes phosphorothioate backbone modifications. (**B**) Timeline of the animal study over a 22-day period. A total of 30 female BALB/c mice (6–8 weeks old) were randomly divided into six groups. Ovalbumin (OVA)-induced allergy was established by subcutaneous sensitization with 20 µg OVA on Days 1 and 8, followed by a challenge with 200 µg OVA on Day 21. Group B received OVA alone and served as the allergic control. Groups C–F were co-treated with OVA and one of the four ODNs: Group C (2m’-V3), Group D (3m-V4), Group E (um-V2), and Group F (2m-V1), to assess their immunomodulatory potential under allergic conditions. Groups A (PBS vehicle control) and F (2m-V1, known positive immunomodulator) each contained 3 mice. Group B and selected ODN groups (C–E) included 6 mice, with emphasis placed on comparisons involving unmethylated um-V2, methylated 2m’-V3, and 3m-V4. Clinical allergy symptom scores were assessed 1 h post-OVA challenge. Spleen, serum, and skin tissues at the injection sites were collected on Day 22 following euthanasia.

**Figure 2 ijms-26-06755-f002:**
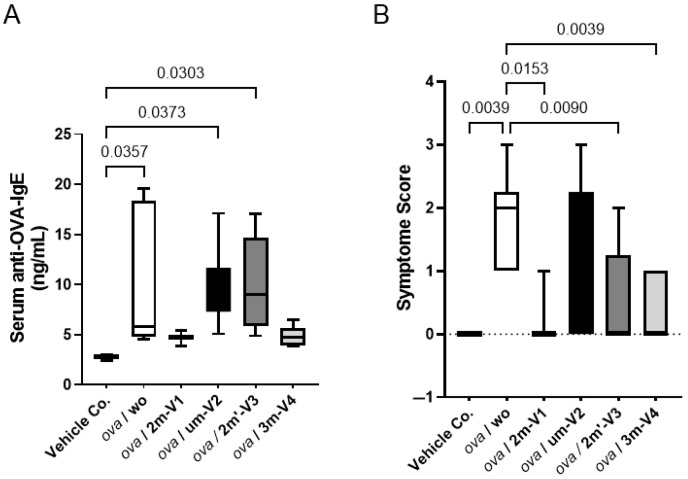
Serum anti-OVA IgE concentrations and clinical allergy symptom scores. (**A**) Serum levels of OVA-specific IgE (ng/mL) were quantified by ELISA from samples collected on Day 22. All 30 mouse serum samples were analyzed in duplicate. Bracket bars indicate pairwise group comparisons. The vehicle control group received PBS and AddaVax adjuvant. “ova/wo” denotes the allergic group treated with OVA alone, while “ova/ODNs” represents groups co-treated with OVA and one of the four ODNs. Notably, methylated 3m-V4 and 2m-V1 significantly reduced serum IgE levels compared to OVA-only controls, whereas unmethylated um-V2 (ova/um-V2) and alternatively methylated 2m’-V3 (ova/2m’-V3) failed to suppress IgE production. (**B**) Median clinical symptom scores were assessed 1 h after OVA challenge for each group. OVA-induced allergic symptoms were markedly alleviated by co-treatments with methylated 2m-V1, 2m’-V3, and 3m-V4, but were unaffected by unmethylated um-V2. Statistical significance (*p* < 0.05) is indicated for relevant group comparisons.

**Figure 3 ijms-26-06755-f003:**
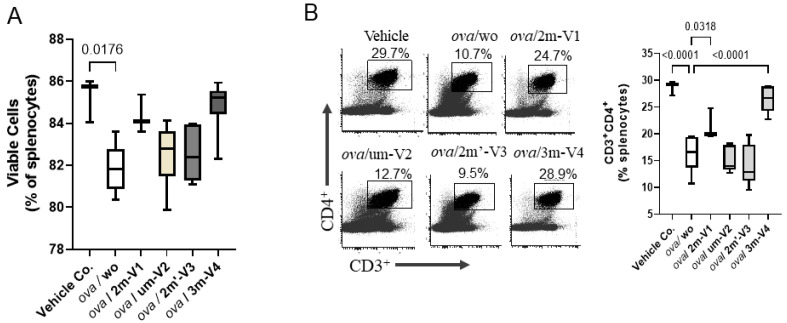
OVA challenge suppresses CD4^+^ T cells and reduces splenic cell viability. BALB/c mice (n = 3–6 per group) underwent a 3-week treatment regimen as outlined in Figure 1B, followed by spleen harvest on Day 22. Splenocytes were isolated and analyzed via flow cytometry. Gating strategies for identifying conventional CD4^+^ T cells and subsets are provided in Appendix A. (**A**) OVA challenge (ova/wo group) significantly reduced the number of viable splenocytes compared to vehicle control. Co-treatment with methylated 2m-V1 or 3m-V4 partially or fully restored splenic cell viability. (**B**) Representative flow cytometry plots show the percentage of CD3^+^CD4^+^ T helper cells in each treatment group. OVA-induced depletion of CD4^+^ T cells was notably prevented by co-treatment with 2m-V1 or 3m-V4, resulting in levels comparable to vehicle controls. Bar graph displays median group values, with statistical comparisons performed by one-way ANOVA. Significant differences (*p* < 0.05) are indicated; non-significant comparisons are unlabeled.

**Figure 4 ijms-26-06755-f004:**
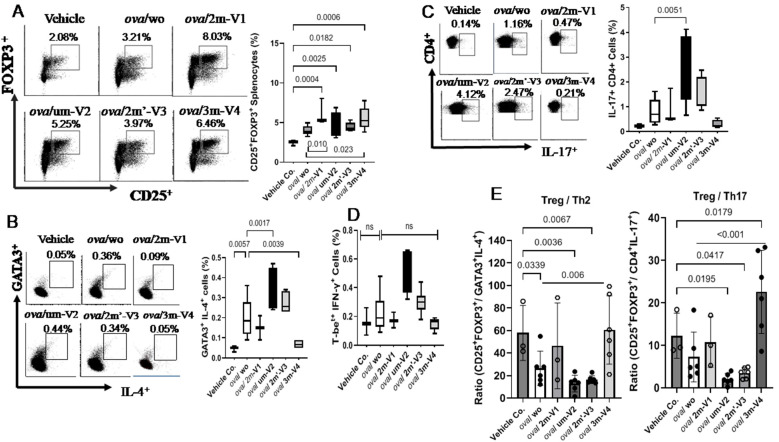
Flow cytometric analysis of T helper cell subset responses in mouse splenocytes from different treated groups. (**A**) The Treg subset (CD4^+^CD25^+^FOXP3^+^) was elevated in all OVA-treated groups compared to vehicle controls. Co-treatment with methylated 3M-V4 or 2M-V1 further enhanced Treg expansion. (**B**) The Th2 subset (CD4^+^IL-4^+^GATA3^+^) was significantly increased in OVA-challenged mice, with an even greater increase observed in the ova/um-V2 group. This Th2 expansion was effectively reduced in mice co-treated with 3m-V4. (**C**) A slight increase in Th17 cells (CD4^+^IL-17^+^) was observed in OVA-only mice, which was significantly amplified in the OVA + um-V2 group (*p* = 0.005). No significant changes were detected with other ODN treatments. (**D**) No significant differences were observed among groups in the Th1 subset (CD4^+^IFN-γ^+^T-bet^+^). Bar graphs next to each flow plot show median values for each group. (**E**) The ratios of Treg to Th2 and Th17 cells were calculated and compared across experimental groups. Data are presented as mean ± SEM. Statistical comparisons were performed using one-way ANOVA; significant differences (*p* < 0.05) are indicated, while non-significant comparisons are not labeled.

**Figure 5 ijms-26-06755-f005:**
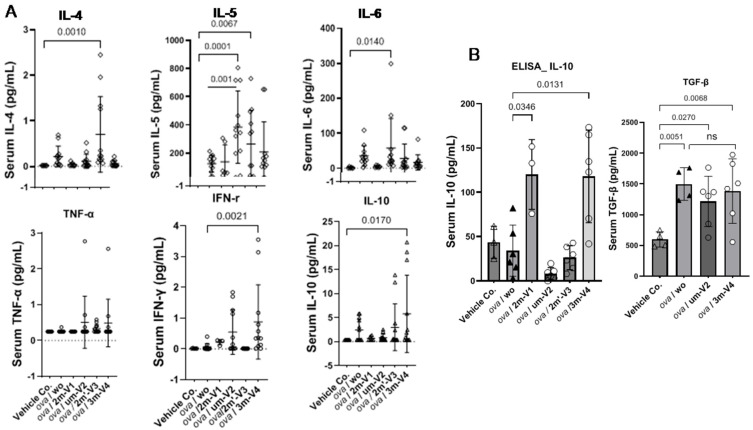
Serum cytokine responses to OVA and CpG ODN treatments. Cytokine levels were quantified using the LEGENDplex™ Multi-Analyte Flow Assay Kit (**A**) and ELISA assays (**B**). OVA treatment alone induced a heightened pro-inflammatory cytokine environment. Among the ODN co-treated groups, two distinct cytokine response patterns were observed. Unmethylated UM-V2 and alternatively methylated 2M’-V3 further amplified Th2-associated cytokines, including IL-4, IL-5, and Th17-associated IL-6, consistent with an exacerbated allergic profile. In contrast, co-treatment with methylated 3M-V4 or 2M-V1 shifted the cytokine milieu toward an immunoregulatory or tolerogenic state, as evidenced by elevated IL-10 levels. Statistical comparisons were performed using one-way ANOVA; significant differences (*p* < 0.05) are indicated, while non-significant comparisons are not labeled.

**Figure 6 ijms-26-06755-f006:**
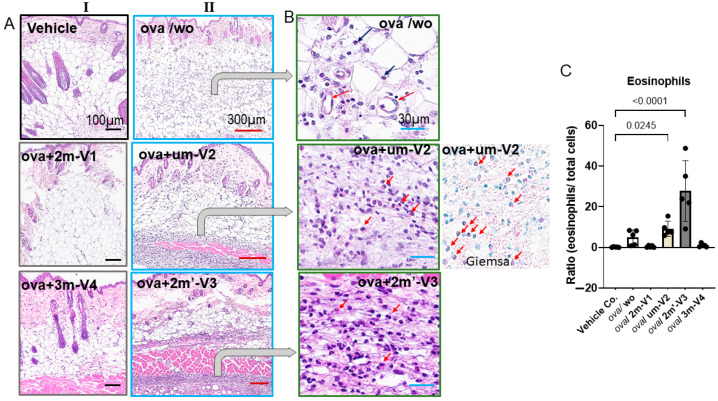
Histopathological evidence that methylated 2m-V1 and 3m-V4 alleviate OVA-induced skin inflammation. (**A**) Representative hematoxylin and eosin (H&E) images of skin tissue sections from each treatment group. In column I, light or no inflammatory response is seen in ova/2m-V1 and ova/3m-V4. Prominent inflammatory cell infiltrates are observed in the dermis and subcutaneous adipose layers of OVA-challenged mice (ova/wo), as well as in ova/um-V2 and ova/2m’-V3 groups (II column). While ova/wo mice exhibit more diffuse infiltration, the presence of um-V2 and 2m’-V3 results in a denser and more band-like pattern of infiltrates. (**B**) Enlarged views of H&E and Giemsa-stained sections highlight abundant eosinophil infiltration (indicated by red arrows) in ova/um-V2 and ova/2m’-V3 mice, characterized by red granular cytoplasm. In contrast, infiltrates in the OVA-only group are predominantly mononuclear cells (black arrows) localized around dilated capillaries. Scale bars are indicated at the top of each image column. (**C**) Quantitative comparison of eosinophil proportions within the inflammatory infiltrate across groups. The OVA + um-V2 and OVA + 2m’-V3 groups show significantly higher eosinophil recruitment, likely driven by their elevated IL-5 response.

**Figure 7 ijms-26-06755-f007:**
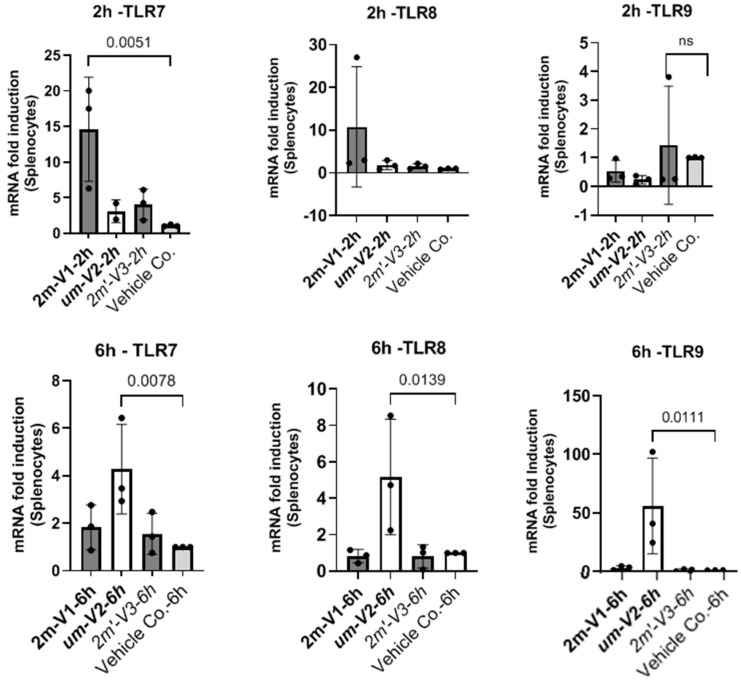
Differential expression of nucleic-acid-sensing TLRs in mouse splenocytes challenged with methylated and unmethylated ODNs in vitro. Mouse splenocytes were cultured in vitro and treated with 25 µg of unmethylated UM-V2, methylated 2M-V1, or alternatively methylated 2M’-V3. Total RNA was extracted at 2 h (top panel) and 6 h (bottom panel) post-treatment, reverse transcribed to cDNA, and analyzed by RT-qPCR for the expression of TLR-7, TLR-8, and TLR-9. Gene expression levels were calculated as fold changes relative to the PBS control after normalization to the ΔCt of internal GAPDH gene. Bar plots display mean fold changes with statistical comparisons performed using one-way ANOVA. Significant differences (*p* < 0.05) are indicated; non-significant comparisons are not labeled.

## Data Availability

The original contributions presented in this study are included in the article and Appendix A. Further inquiries can be directed to the corresponding author.

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
