# Peer review of "Methylated CpG ODNs from Bifidobacterium longum subsp. infantis Modulate Treg Induction and Suppress Allergic Response in a Murine Model"

_ijms, 2025, doi:10.3390/ijms26146755_

Round 1

Reviewer 1 Report

Comments and Suggestions for Authors

This study explores the immunomodulatory effects of methylated CpG ODNs in an allergic mouse model. The topic is relevant, and the findings are interesting. However, several aspects of the study design and data interpretation require clarification or improvement. Please see detailed comments below.

  1. Line 128–132: Please clarify why only 3 mice were included in Groups A and F, while the other groups included 6 mice. The small sample size in all groups, particularly in Groups A and F, limits the statistical power of the study and may affect the reliability of the conclusions.
  2. 2.2 Animals and groups: While PBS and OVA-alone groups were included as controls, no non-CpG or scrambled ODN control group was used. This makes it difficult to determine whether the observed immunomodulatory effects are specifically due to the CpG motif itself. It is recommended to include one or more non-functional ODN controls to rule out non-specific sequence effects or immune activation caused by the phosphorothioate backbone alone.
  3. TLR7/8/9 expression was assessed in vitro, but no in vivo validation was provided. Consider adding functional experiments (e.g., TLR9 antagonist or knockdown) to confirm whether the effects of ODNs are TLR-dependent.

Author Response

We thank the reviewer for your careful evaluation and constructive suggestions. Below is our one-by-one responses for the comments. 

  1. Line 128–132: Please clarify why only 3 mice were included in Groups A and F, while the other groups included 6 mice. The small sample size in all groups, particularly in Groups A and F, limits the statistical power of the study and may affect the reliability of the conclusions.

Response: We thank the reviewer for this important observation. Groups A and F included only three animals each in order to minimize overall animal use, consistent with ethical guidelines, as these served as well-established controls validated in our previous experiments (Ref. 14). In contrast, Groups C and D tested novel methylated ODN formulations (2m'-V3 and 3m-V4) for the first time, warranting larger group sizes (n=6) to better assess variability and treatment effects.

      Although the immune-modulatory effect of Group E (unmethylated um-V2 ODN) was also demonstrated in our earlier study, we included six animals in this group in the present study to ensure repeatability and consistency of the results. Importantly, the unmethylated um-V2 group also served as a critical contrast group for evaluating the effects of the two newly designed methylated ODNs in the current study. Notably, Groups A, F, and E again showed the expected outcomes in this experiment, which increases our confidence in the relative effectiveness of the new methylated ODN treatments.

  1. 2.2 Animals and groups: While PBS and OVA-alone groups were included as controls, no non-CpG or scrambled ODN control group was used. This makes it difficult to determine whether the observed immunomodulatory effects are specifically due to the CpG motif itself. It is recommended to include one or more non-functional ODN controls to rule out non-specific sequence effects or immune activation caused by the phosphorothioate backbone alone.

Response: We appreciate this insightful suggestion. We acknowledge that our current experimental design did not include a non-CpG or scrambled ODN control group, which would indeed help to more definitively exclude nonspecific immunostimulatory effects from the phosphorothioate backbone. We agree this is an important consideration, and we plan to include such non-functional ODN controls in future studies to strengthen mechanistic conclusions.

      However, we would like to note that all ODNs used in this study shared the phosphorothioate backbone, reducing the likelihood that the observed immunological differences are attributable to the backbone itself. Notably, the 2m'-V3 group (a methylated ODN variant) showed effects comparable to the unmethylated um-V2 group—both lacking the Treg induction and IL-10 upregulation observed in the other two methylated ODN groups (3m-V4 and 2m-V1). This outcome suggests that 2m'-V3, despite its methylation and backbone structure, may be functionally inactive in this context.

     Therefore, 2m'-V3 may serve as a relevant non-functional methylated ODN control in future studies. While we acknowledge that it does not replace a formal non-CpG or scrambled sequence control, these results strengthen our confidence that the observed immune modulation is not due to nonspecific effects of the backbone alone.

  1. TLR7/8/9 expression was assessed in vitro, but no in vivo validation was provided. Consider adding functional experiments (e.g., TLR9 antagonist or knockdown) to confirm whether the effects of ODNs are TLR-dependent.

Response: We thank the reviewer for this thoughtful and important suggestion. We agree that using TLR9 antagonists or knockdown approaches would provide valuable mechanistic confirmation of TLR dependency. However, our current in vivo design focused on assessing downstream immunological outcomes rather than the immediate pathogen recognition stage. Our sample collection occurred on Day 22 (24 hours after the third ODN administration), a timepoint likely beyond the initial PAMP–PRR interaction window. Thus, the immune responses measured in vivo reflect downstream consequences of early signaling events, making direct in vivo TLR antagonism less informative in this context.

Additionally, while TLR9 has been classically implicated in sensing unmethylated CpG motifs, the mechanism of immune sensing for methylated CpG ODNs remains less well defined. Our in vitro data showed elevated TLR7 expression specifically in the methylated 2m-V1 group at 2 hours, suggesting a possible role for TLR7 in recognizing certain methylated ODN variants. Moreover, as we discussed in the manuscript, non-TLR sensing pathways such as the cGAS–STING axis may also contribute to the observed effects.

We recognize the importance of dissecting these pathways more precisely. We have already attempted in vitro experiments using DC differentiation from THP-1 and mouse bone marrow cells to study receptor involvement, but encountered technical challenges in generating sufficient DC yields for analysis. We recognize the need to dissect these pathways more rigorously. In future work, we plan to address these limitations and expand our approach to include experiments with TLR9 agonists and antagonists, as well as to investigate TLR7 and non-TLR pathways in pDC-based mechanistic studies.

Reviewer 2 Report

Comments and Suggestions for Authors

The research article titled “Structure-function and Treg induction of methylated CpG ODNs from Bifidobacterium longum susp infantis in an allergic murine model” authored by Li D et al, studies the different immunomodulatory aspects of CpG-ODN and how methylation can affect the immunological outcomes. They have used different methylated forms of the CpG-ODN derived from B.longum in adjunction with OVA in a murine allergy model to study their ability to skew the immune landscape. They conclude that specific methylation modifications of the CpG-ODN can lead to suppression of a pro-Type 2 immune response to a more tolerogenic response by induction of Tregs and elevation of IL-10. The study could benefit by considering the following suggestions/comments:

Major comments:

  1. The title could be worded better. It is unclear by the term “structure-function”. The title could instead read “Methylated CpG ODNs from Bifidobacterium longum susp infantis modulate Treg induction in a murine allergy model”.
  2. Lines 40 -41 are plagiarized from Montamat G et al, Front. Immunolo, 2021.
  3. The introduction can include more recent studies and reviews for example, Wang Y et al, Front immuno, Jan 2024, Lin YJ et al, Front immune, Feb 2024.
  4. There are a lot of grammatical errors from Lines 110-113. Please reword these sentences.
  5. Mouse numbers used per group are not enough since there is a lot of spread in the data. The number of repeats of the experiments are not sufficient either. There is a visible trend in data, however, statistical significance is lacking which is not convincing enough to accept the data in the present form.
  6. The authors investigate the T-cell compartment into great detail; however, it is known that pDCs drive the induction of Treg population to create a tolerogenic environment. The authors could have investigated the pDC compartment to add rigor to the study.
  7. Similarly, the authors do not show the status of the B-cell compartment even though they use markers for B-cells in the methods. Authors must include this in the study.

Minor comments:

  1. Please fix methods heading in line 200.
  2. In Fig 2, please keep the multiple comparisons consistent between the 2 graphs to better understand the findings.
  3. The authors can show the OVA specific IgE level of the groups post 14/15 days of sensitization phase. Will it be elevated compared to the vehicle group to suggest sensitization in the um-V2 group?
  4. In figure legend for Fig 3, the authors mention cotreatment with 2m-V1 or 3m-V4 results in CD4 T cell numbers restoration comparable to vehicle controls, but the comparison is not shown in the graph in Fig 3B. Please show this comparison in the graph.
  5. In Fig 6, it is good to show both column I and II figures in the same magnification, preferably 300um and show similar orientation of the tissues to be able to see the submucosa in all the groups.
  6. Line 444, the figure reference should be Fig 6B and in line 449, figure reference should be 6C.
  7. Fig 7 is not referenced anywhere in the text. Please include it in text section 3.6.
  8. In Fig 7, 6h-TLR9 graph, the 3rd group mentioned in the graph is wrong. It should be 2m’-V3-6h unless otherwise mentioned in the text.
  9. As mentioned above in the major comment #7, B cell data is lacking even though markers to identify the same are mentioned in the materials. The supplementary Figure S1 shows gating strategy to identify the different T cell populations however, B cells identification is not shown. Please address this. 

Author Response

We thank the reviewer for their careful evaluation and constructive suggestions, which have helped us improve the quality and clarity of our manuscript.

Major comments:

  1. The title could be worded better. It is unclear by the term “structure-function”. The title could instead read “Methylated CpG ODNs from Bifidobacterium longum susp infantis modulate Treg induction in a murine allergy model”.

Response: We thank the reviewer for suggesting improvements to the title for greater clarity. We agree that “structure-function” was too vague and have revised the title to better reflect the main outcomes of this study. Our proposed new title is:“Methylated CpG ODNs from Bifidobacterium longum subsp. infantis modulate Treg induction and suppress allergic responses in a murine model.”
We believe this title revision clearly conveys both the mechanistic aspect (Treg modulation) and the main outcome (suppression of allergic responses).

  1. Lines 40 -41 are plagiarized from Montamat G et al, Front. Immunolo, 2021.

Response: We agree that the sentence in question was too similar in wording. We have revised it to avoid any potential plagiarism while preserving the intended meaning. The revised sentence now reads: Among treatments for allergic diseases, allergen-specific immunotherapy uniquely offers long-term efficacy by modifying the underlying immune response.” We appreciate the reviewer’s careful attention to this detail.

  1. The introduction can include more recent studies and reviews for example, Wang Y et al, Front immuno, Jan 2024, Lin YJ et al, Front immune, Feb 2024.

Response: Three references have been added in the introduction as 6, 7, and 8.

  1. There are a lot of grammatical errors from Lines 110-113. Please reword these sentences.

Response: Thanks for reviewer’s careful attention to this detail. The paragraph has been revised and highlighted in yellow.

  1. Mouse numbers used per group are not enough since there is a lot of spread in the data. The number of repeats of the experiments are not sufficient either. There is a visible trend in data, however, statistical significance is lacking which is not convincing enough to accept the data in the present form.

Response: We appreciate the reviewer’s concern about statistical power and the observed variability in the data. We agree that ~10 mice per group would ideally provide greater statistical confidence. Since this experiment is part of an ongoing series of studies investigating methylated ODN immune responses both in the presence and absence of allergen, we would like reduce overall animal use while ensuring meaningful comparisons. We used fewer mice in Groups A and F (n=3), which serve as well-established controls repeatedly validated in our prior work [17,18]. For the novel methylated ODN formulations tested in this study (Groups C and D), we used larger group sizes (n=6) to better capture variability and evaluate treatment effects. Similarly, although the immunostimulatory effects of the unmethylated um-V2 ODN (Group E) have been consistently observed in our previous studies, we included 6 mice in this group to confirm repeatability and allow balanced comparison with the new methylated formulations.

We have clarified these sample size considerations in the manuscript with the following added paragraph:

Groups A and F comprised fewer animals (n=3) to reduce overall animal use, as they have served as well-established controls in our previous experiments [17,18]. By contrast, Groups C and D involved novel methylated ODN formulations (2m'-V3 and 3m-V4) and required larger group sizes (n=6) to evaluate variability and treatment effects. Group E also included 6 mice to ensure that the previously observed immune-stimulatory effects of the unmethylated um-V2 ODN were consistently reproduced, enabling more robust comparison to the new treatments.

  1. The authors investigate the T-cell compartment into great detail; however, it is known that pDCs drive the induction of Treg population to create a tolerogenic environment. The authors could have investigated the pDC compartment to add rigor to the study.

Response: We thank the reviewer for this valuable suggestion. While our current in vivo design focused on assessing downstream immunological outcomes rather than the early pathogen-recognition stages, we fully agree on the critical role of pDCs in driving Treg induction and creating a tolerogenic environment. This is an important mechanistic question we are actively pursuing. In fact, we have already attempted in vitro experiments using DC differentiation from THP-1 and mouse bone marrow cells to study receptor involvement, but encountered technical challenges in generating sufficient DC yields for analysis. We recognize the need to dissect these pathways more rigorously. In future work, we plan to address these limitations and expand our approach to include experiments with TLR9 agonists and antagonists, as well as to investigate TLR7 and non-TLR pathways in pDC-based mechanistic studies.

  1. Similarly, the authors do not show the status of the B-cell compartment even though they use markers for B-cells in the methods. Authors must include this in the study.

Response:  We thank the reviewer for this important comment. We have indeed evaluated B-cell markers in our previous studies (Refs. 17, 18), where we specifically examined Breg populations (e.g., FoxP3⁺ or IL-10⁺ B cells). However, across multiple experiments, we found that Breg frequencies did not show consistent or significant changes under the various treatment conditions—regardless of whether IgE responses were elevated or suppressed. Based on these findings, we chose not to include detailed B-cell phenotyping in the current study, in order to focus on the more relevant T-cell and myeloid cell responses. Moreover, given the evidence for non-TLR9 mechanisms in responses to methylated ODNs observed here, we suspect that classical TLR9-dependent B-cell activation is less likely to be a major driver in this context. Instead, we hypothesize that pDC-mediated regulatory mechanisms may play a more important role, which we plan to investigate further in future mechanistic studies.

 Minor comments:

  1. Please fix methods heading in line 200.

Response: corrected

  1. In Fig 2, please keep the multiple comparisons consistent between the 2 graphs to better understand the findings.

Response: Thank you for this helpful suggestion. The apparent inconsistency in the multiple comparisons arises because in Figure 2A we did not label non-significant (n.s.) comparisons explicitly in the graph. In Figure 2A, although the OVA-alone group shows substantial variability in serum IgE levels, the unmethylated ODN and the alternative methylated ODN groups both produced intermediate IgE levels, while the two primary methylated ODN groups showed lower IgE responses that were still significantly different from the vehicle control. 

  1. The authors can show the OVA specific IgE level of the groups post 14/15 days of sensitization phase. Will it be elevated compared to the vehicle group to suggest sensitization in the um-V2 group?

Response: We are not sure the question. In our previous study (Ref. 17) and in the current work, we observed clear signs of sensitization in the um-V2 group, reflected by elevated OVA-specific IgE levels and other Th2-associated responses. However, it is important to note that in this study, all relevant groups received a mixture of OVA + ODNs, making the adjuvant role of the ODNs central to interpreting the OVA induced IgE response. In our earlier work, when um-V2 was administered alone without OVA, none of the 6 mice showed elevated OVA-specific IgE. This supports the idea that the ODN itself does not act as an antigen but rather functions as an adjuvant that can enhance or skew the allergic inflammatory response when co-administered with the allergen. This provides a rationale for developing an AIT-ODN–type vaccine approach.

  1. In figure legend for Fig 3, the authors mention cotreatment with 2m-V1 or 3m-V4 results in CD4 T cell numbers estoration comparable to vehicle controls, but the comparison is not shown in the graph in Fig 3B. Please show this comparison in the graph.

Response: We did show a bar graph next to flow plots on the right side.  Is it possible this was cut off when you printed it out?

  1. In Fig 6, it is good to show both column I and II figures in the same magnification, preferably 300um and show similar orientation of the tissues to be able to see the submucosa in all the groups.

Response: the reviewer is not wrong here, but we prefer to make clear the existence of an inflammatory infiltrate in col. 2 at whatever scale we are obliged to use so that the reader can see it (otherwise the reader may worry that there is a null result in the data).  However, we have included words to this effect in the legend, so that the reader is not led astray.

  1. Line 444, the figure reference should be Fig 6B and in line 449, figure reference should be 6C.

Response: Thanks for pointing these out. Both figure references were corrected.

  1. Fig 7 is not referenced anywhere in the text. Please include it in text section 3.6.

Response: Thanks for your pointing this out. Figure 7 has been referenced in the text section 3.6.

  1. In Fig 7, 6h-TLR9 graph, the 3rdgroup mentioned in the graph lacking even though markers to identify the same are mentioned in the materials.

Response:  Thanks for pointing this out, we have revised the third group label in 6h-TLR9 graph.

  1. As mentioned above in the major comment #7, B cell data is lacking even though markers to identify the same are mentioned in the materials. The supplementary Figure S1 shows gating strategy to identify the different T cell populations however, B cells identification is not shown. Please address this. 

Response: Based on negative findings in previous study, we chose not to include detailed B-cell phenotyping in the current study, in order to focus on the more relevant T-cell and myeloid cell responses. We have revised the method section to remove any material related to B cells identified by CD45R/B220 (clone RA3-6B2).

Round 2

Reviewer 1 Report

Comments and Suggestions for Authors

The authors have addressed all of my previous comments and revised the manuscript accordingly. I have no further concerns.

Author Response

Comments: The authors have addressed all of my previous comments and revised the manuscript accordingly. I have no further concerns.

Response: Thanks for your time and effort. They are much  appreciated. 

Dongmei Li

Reviewer 2 Report

Comments and Suggestions for Authors

Line 200, section 2.8 heading not fixed yet in the new version. 

Author Response

Comments: Line 200, section 2.8 heading not fixed yet in the new version. 

Response:  Thanks for your time and effort. They are appreciated.  The section 2.8 heading has now been fixed.